# Development of social isolation and social network assessment tool for older adults: A Delphi survey

**Ah-Ram Kim[1¤‡], Kang-Hyun Park[2]\***

**1** Department of Occupational Therapy, College of Health and Medical Services, Sangji University, Wonju, South Korea, **2** Department of Occupational Therapy, Baekseok University, Cheonan, South Korea

‡ This author share first author on this work.
¤ Current Address: Department of Occupational Therapy, Sangji University, Wonju, South Korea
* kanghyun@bu.ac.kr

## Abstract

### Objective

Social Isolation and network are crucial factors which affect the quality of life and overall health of older adults. Therefore, in the field of health promotion, there's a growing attention to assess and evaluate the social health including social isolation and network of older adults. This study aimed to develop a Social Isolation and Social Network (SISN) evaluation tool that comprehensively measure the social isolation and network of older adults.

### Methods

Between April and June 2024, we gathered insights from 23 experts in lifestyle and health. Based on the previous research, a two-round Delphi survey was conducted to obtain expert consensus on key assessment components. In the initial survey, we collected expert's opinions through open and closed-ended questions about social isolation and social network evaluation items. After the first survey, we corrected several items that didn't meet the Content Validity Ratio (CVR) standard before proceeding. We presented the first survey's findings to an expert panel, leading to a consensus.

### Results

The initial Delphi round produced 32 items and the second round resulted in 30 items after adjusting those with CVR scores of 0.37 or less. The final CVR was 0.87 with a convergence of 0.87, a consensus level of 0.31 and stability level of 0.12.

**Data availability statement:** All relevant data are within the manuscript.

**Funding:** This work was supported by the National Research Foundation of Korea grant funded by the Korean Government (RS-2023-00213828). The funders had no role in study design, data collection and analysis, decision to publish, or preparation of the manuscript.

**Competing interests:** The authors have declared that no competing interests exist.

## Conclusion

This study successfully developed a comprehensive social isolation and social network assessment using a modified Delphi technique. The 30 items are categorized into three sections. The SISN is able to provide a systematic approach to evaluate social isolation and social network of older adults. Future studies should be conducted to examine the reliability and validity.

## Introduction

Social isolation and networks significantly affect the quality of life and overall health of older adults [1]. Older adults can become vulnerable to social isolation owing to various factors, including retirement, spousal bereavement, and the death of friends [2]. Social isolation is a state in which an individual lacks social contact or interaction with others [3]. This can manifest in different forms, including physical isolation (being physically separated from others) and perceived isolation (feeling lonely or disconnected even when surrounded by others) [4]. Berkman and Syme [5] first identified social isolation as a risk factor in their study on social relationships and mortality, which has since been continuously reported in the field of older adult health.

Social isolation is directly linked to various health problems, such as depression, anxiety, and cognitive decline [6], which may eventually increase the risk of premature death [7]. Conversely, social networks play a crucial role in improving the health and well-being of older adults by providing emotional support, information, and opportunities for social participation [8]. Previous studies have emphasized the importance of social isolation and networks. According to Valtorta et al. [9], social isolation is a major factor that increases the risk of cardiovascular disease and mortality. Additionally, the size and diversity of social networks positively affect health [10,11]. However, these studies relied mainly on quantitative data and did not sufficiently reflect the qualitative aspects of social isolation and networks.

Existing tools for assessing social isolation and networks in older adults have several limitations. Firstly, these tools tend to rely primarily on quantitative approaches to measure the number and frequency of relationships [12]. Thus, there are limitations to fully understanding social isolation of older adults through simple quantitative measurements. For instance, the Lubben Social Network Scale (LSNS), which has been widely used to assess social integration and social isolation among community-dwelling populations [13], has some constraints. Although the LSNS instrument demonstrates sound psychometric properties and is easy to administer [14], it relies solely on self-administered questions and lacks a gold standard [13]. Similarly, the Social Network Index (SNI), a self-reported measure of social ties based on an individual's number of social connections, also tends to focus on the quantity and frequency of relationships while lacking measurement of emotional aspects. For example, even if individuals have contact with many people, they may still feel isolated if they lack deep emotional bonds [15]. Furthermore, it is essential to evaluate the qualitative aspects of relationships, such as relationship satisfaction, depth of emotional

support, and quality of interactions. Tools that do not sufficiently incorporate these qualitative aspects fail to accurately represent the actual experiences and needs of older adults, potentially hindering the effectiveness of policymaking and intervention programs. Therefore, there is an urgent need to develop more comprehensive and reliable evaluation tools that reflect older adults' actual experiences and needs.

This study aimed to develop a new, more comprehensive evaluation tool through expert consensus using the Delphi technique, the Social Isolation and Social Network (SISN) evaluation tool, to comprehensively assess the social isolation and networks of older adults. To achieve this, the following research questions were formulated: (1) What items are necessary to evaluate social isolation and networks of older adults? (2) What is the reliability and validity of the evaluation items agreed upon by experts? (3) What are the advantages and limitations of the newly developed evaluation tool compared with existing evaluation tools?

This study was conducted by collecting opinions from expert groups and achieving consensus using the Delphi technique [16]. This approach allows for a more detailed evaluation of social isolation and networks of older adults. Additionally, the SISN tool can be applied to various fields related to the welfare of older adults and can provide essential data for policymaking and program development.

## Methods

### Research design

This study employed a modified Delphi survey technique [17] which is a flexible approach for gaining experts consensus through iterative round and controlled feedback [18]. Unlike in the traditional Delphi method, our modified approach initiated the first survey was conducted during the second round; the validity of the evaluation indicators can be verified and supplementary opinions can be presented using the applied structured questionnaire. The modified Delphi survey incorporated both open- and closed-ended questions. Open-ended questions were used to gather additional insights and suggestions that informed the construction of closed-ended questions for subsequent rounds. This study was approved by the Baekseok University Institutional Review Board (approval number 24HR02), and informed consent was obtained from all participants.

### The expert panel

We included multidisciplinary experts, including occupational therapists, physical therapists, nurses, and social workers. Experts were selected based on their extensive experience and expertise in research and clinical fields related to social isolation and networks. The panel consisted of experts with over five years of experience in Korea, ensuring a comprehensive and holistic approach to the development of a social isolation and network evaluation questionnaire. The inclusion criteria were: (1) unrestricted access to the Internet and email; (2) proficiency in using computers for documentation; and (3) the ability to complete two Delphi surveys within two months. Before initiating the Delphi survey, an explanation of the study and consent forms were prepared and sent to all prospective panelists via email. Only participants who provided written informed consent were included in the final panel. After obtaining consent for participation, a Delphi survey was conducted. For those selected as potential experts, an invitation email was sent that included a letter introducing the Delphi survey, a consent form related to the study, and an IRB guide. The experts were informed that they had to respond to all rounds until a consensus was reached, each round was given an appropriate amount of time to respond, participation was voluntary, and they could withdraw at any time. They were also informed that their individual responses would not be shared with other expert panels and their contributions would be kept confidential. Delphi surveys were conducted only with experts who expressed consent, and as a reward for their time, participants were given a $50 gift card after completing all the Delphi surveys.

### The Delphi method

The study was conducted with participants who provided consent to potential expert panels. Demographic information was collected from the participants. Participants were instructed to read the guidelines introducing the concept of social

isolation and social network developed to create SISN assessment tools in the introduction of the survey attached to the email. The first section explains the social isolation and social network factors based on a literature review and previous studies. Participants were then shown the following sections consisting of 35 questions: 7 items of objective social isolation, 10 items of subjective social isolation, 15 items of social and 3 open-ended questions. These items were developed based on literature review.

The Delphi survey consisted of three steps (Fig 1). In the first stage, a literature review of the development of the questions was derived from previous studies. In the second stage, the Delphi was conducted with experts, and the Delphi had a total of two rounds. The research team provided information about the study and obtained consent from all participants from a panel of experts before the study began. The participants responded to the first- and second-round questionnaires delivered through e-mail. Each round of the survey was conducted for one week, and to maximize the response rate, participants who did not respond were given additional time to send a reminder.

## Round 1

The survey was open and closed to gather the opinions of the expert panel. The first survey consisted of 32 closed-ended and three open-ended questions. Each participant was encouraged to submit their revised recommendations and additional comments regarding the questionnaire. Round 1 of the survey took approximately 35 minutes to complete.

## Round 2

The Round 2 survey was revised to reflect the responses of the expert panels in Round 1. It consisted of 30 closed-ended questions in three domains (objective isolation, subjective isolation and social network). Each participant was informed via e-mail and asked to answer each question using a 5-point Likert scale ranging from 1(strongly irrelevant) to 5 (strongly agree). In the second round, four items from the first round were modified.

## Data analysis

In the first round, the panelists provided their opinions on the importance and suitability of each evaluation item and freely submitted additional opinions through open-ended questions. Unclear expressions or questions were revised and

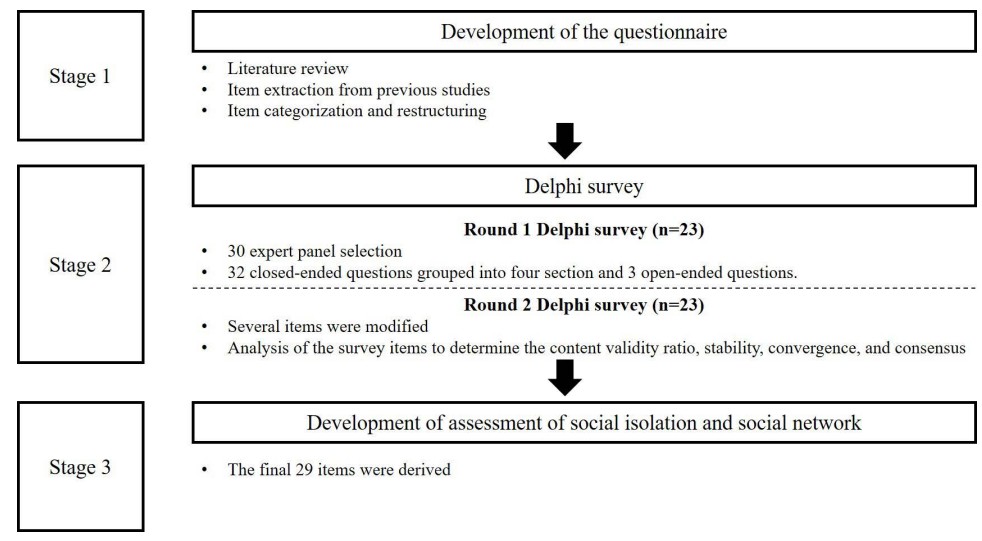

**Fig 1. Research process.**

supplemented based on panelists' feedback. The content validity ratio (CVR), stability, convergence, and consensus were used to analyze the results.

The reference value of the CVR was determined based on the number of panelists participating in each round. The CVR is an indicator of the significance assigned to a particular item by a group of experts, and it assesses the content validity of the item. The formula for calculating the CVR value is as follow [19]:

$$CVR = \frac{ne - \frac{N}{2}}{\frac{N}{2}}$$

The *ne* indicates number of panelists responding with 4 points or more (I agree) and N indicates number of total panelists. The CVR value range from -1.0 to +1.0, with a positive number indicating that most of the panelists indicated "I agree" (4 or 5 points). For the fitness value of each item, the content validity ratio (CVR) value was analyzed based on the criteria of Lawshe [19], and the data were analyzed based on 0.37, the minimum value according to the 23 expert panels participating in this study. The convergence, which indicates the degree of opinion convergence among panelists, was calculated using the interquartile range (Q3-Q1)/2, where Q3 and Q1 represent the third and first quartiles, respectively. A lower convergence indicates greater consensus among expert opinions. Given our use of a 5-point Likert scale, we established a convergence criterion of 0.50 or less as indicating acceptable convergence [20]. The formula for calculating the CVR value is as follow:

$$Convergence = \frac{Q3 - Q1}{2}$$

Consensus refers to the degree of agreement on specific items within a group; the higher the consensus, the higher the agreement among experts [21]. The consensus index also uses the third quartile (Q3) and first quartile (Q1) in the numerator but differs in that it is divided by the median. The formula is as follows:

$$Consensus = 1 - \frac{Q3 - Q1}{Mean}$$

## Results

### Demographics of the expert panels

Thirty experts participated in this study, and 23 (73.3%) completed the questionnaire. The recruitment flow and study cohort size per round are shown in Fig 1. The majority of participants (52.17%) had clinical experience of 10–19 years. This study's expert panel represented a multidisciplinary composition. The panel consisted predominantly of occupational therapists (70%, n = 16), with additional representation from physical therapists (17%), nurses (9%), and social workers (4%). Their demographic characteristics are presented in Table 1.

### Results of round 1

The categories and items of the detailed results of the Delphi Round 1 are presented in Table 2. Considering that the number of participating experts was 23, the evaluation index of round 1 was refined by leaving items with a CVR coefficient of ≥ 0.37 in the final evaluation results [19]. The first round consisted of 35 questions, including 32 closed-ended questions and 3 open-ended questions in three categories. In the objective social isolation category, the range of the CVR ratio was 0.13~1.00. The question number 2 regarding about the current job was 0.13 thus the consensus was not reached. In terms of subjective social isolation category, the question number 4 asking about the feeling regarding missing people was

demonstrated the low level of CVR ratio (0.30). In the case of social networking category, the question number 6 was not reached the standard value of CVR (0.30). As a result of the first Delphi survey, three questions were removed because it was not reached the consensus values and a new question regarding the usage of social network services was added because social networks are actively formed through social network sites (SNSs), the experts suggested that questions regarding the use of SNSs was necessary. Therefore, a new question on SNS use was added in objective social isolation category. After removing and modifying the questions, a second Delphi was conducted. The categories and items of the detailed results of the Delphi Round 1 are presented in Table 2. Twenty-three experts responded to all questions in the first-round survey, and the first round consisted of 35 questions, including 32 closed-ended questions and 3 open-ended questions in three categories.

## Results of Round 2

The detailed results of the Delphi Round 2 are presented in Table 3. Twenty-three experts responded to all the questions in the second survey. In the first round, the questionnaire was administered by modifying three items that did not reach consensus because they did not reach the standard CVR ratio and adding a new item in objective social isolation. As a result of the round 2, in the objective isolation, subjective isolation, and social network, the CVR of sub-items was ≥ 0.37 and the content validity was verified for all items (Table 3). As a result of round 2, compared with round 1, the relevance of the items was relatively high, and the panel's responses resulted in relatively high convergence and agreement (Table 4).

## Discussion

This study aimed to gather multidisciplinary experts to develop an assessment tool that could measure individuals' social isolation and networks. Following the second survey round, the expert panel identified what they considered to be valid components for measuring social isolation and networks. In the first round, 23 experts who agreed to participate completed and returned the questionnaire. After analyzing the responses using predetermined criteria for consensus and convergence, 35 items which include 32-closed ended questions and 3 open-ended questions in three categories were included in the final version of the questionnaire.

Objective isolation is the tangible and quantifiable lack of connections with other people and is normally measured by the lack of contact/interaction with social-network members [22]. However, there are limitations to simply measuring an individual's objective social isolation based on connections with family or friends. In this study, the majority of experts agreed that index which evaluate individual's objective isolation (Table 3). This consensus among our expert panel aligns with previous literature findings. Specifically, previous research has demonstrated that social networks can be broken

**Table 1. Demographics of the expert panels.**

|  |  | Round 1 (%) | Round 2 (%) |
|---|---|---|---|
| Sample |  | 23 | 23 |
| Response rate (%) |  | 100% | 100% |
| Gender | Male | 11 (47.83) | 11 (47.83) |
|  | Female | 12 (52.17) | 12 (52.17) |
| Work experience | 5~9 years | 6 (26.09) | 6 (26.09) |
|  | 10~19 years | 12 (52.17) | 12 (52.17) |
|  | ≥20 years | 5 (21.74) | 5 (21.74) |
| Major field of study | Occupational therapist | 16 (69.57) | 16 (69.57) |
|  | Physical therapist | 4 (17.38) | 4 (17.38) |
|  | Nurse | 2 (8.70) | 2 (8.70) |
|  | Social worker | 1 (4.35) | 1 (4.35) |

**Table 2. Social isolation and social network assessment in the round 1st survey.**

| Category | No. | Sub-items | Mean* | SD** | CVR*** |
|---|---|---|---|---|---|
| Objective isolation | 1 | How many family members or housemates live with you? | 4.78 | 0.41 | 1.00 |
| | 2 | What is your current job? | 3.65 | 1.05 | 0.13 |
| | 3 | What public transportation is available where you currently live? | 4.00 | 1.10 | 0.39 |
| | 4 | Please select whether public transportation is conveniently accessible from your home. | 3.78 | 1.14 | 0.39 |
| | 5 | Please indicate how many times you go out of the house on average per day. | 4.61 | 0.77 | 0.83 |
| | 6 | Do you have acquaintances who visit you every day, and if so, how many? | 4.30 | 0.75 | 0.83 |
| | 7 | How many people call you or ask how you are every day? | 4.57 | 0.77 | 0.83 |
| Subjective isolation | 1 | Over the past month, I've really missed my close friend. | 4.04 | 1.00 | 0.39 |
| | 2 | Over the past month, I have felt a general sense of emptiness. | 4.17 | 1.01 | 0.48 |
| | 3 | Over the past month, I've felt like my circle of friends and acquaintances was too limited. | 3.91 | 1.02 | 0.39 |
| | 4 | For the past month, I've felt like I miss people around me. | 3.96 | 1.00 | 0.30 |
| | 5 | Over the past month, I have felt separated from others. | 4.48 | 0.83 | 0.74 |
| | 6 | Over the past month, I have felt isolated from other people. | 4.43 | 0.82 | 0.74 |
| | 7 | For the past month, I have felt lonely and friendless. | 4.48 | 0.71 | 0.74 |
| | 8 | For the past month, I've felt like I had no one to rely on. | 4.52 | 0.65 | 0.83 |
| | 9 | Over the past month, I've thought to myself that there are enough people around me that I feel close to. | 4.61 | 0.64 | 0.83 |
| | 10 | Over the past month, I have felt "in tune" with the people around me. | 4.04 | 1.04 | 0.48 |
| Social network | 1 | How many family members and relatives do you see or hear from more than once a month? | 4.30 | 1.08 | 0.57 |
| | 2 | How often do you see or hear from the family members and relatives with whom you have the most contact? | 4.04 | 1.00 | 0.39 |
| | 3 | How many of your family and relatives do you feel comfortable enough to talk to about personal matters? | 4.52 | 0.65 | 0.83 |
| | 4 | How many family members and relatives do you feel close enough to ask for help? | 4.70 | 0.55 | 0.91 |
| | 5 | When one of your family members and relatives has to make an important decision, how often does he or she talk about it with you? | 4.09 | 1.10 | 0.39 |
| | 6 | How often can you talk to one of your family members and relatives when you have to make important decisions? | 4.04 | 0.95 | 0.30 |
| | 7 | Is there always someone you can talk to about everyday problems? | 4.74 | 0.61 | 0.83 |
| | 8 | Are there a lot of people you can turn to when you have a problem? | 4.78 | 0.51 | 0.91 |
| | 9 | Are there many people I can completely trust? | 4.57 | 0.71 | 0.74 |
| | 10 | Do I have friends I can call whenever I need them? | 4.48 | 0.71 | 0.74 |
| | 11 | How many friends do you see or hear from more than once a month? | 4.52 | 0.83 | 0.74 |
| | 12 | How often do you see or hear from the friends you have the most contact with? | 3.96 | 1.12 | 0.48 |
| | 13 | How many friends do you have with whom you feel comfortable enough to talk about personal matters? | 4.78 | 0.41 | 1.00 |
| | 14 | How many friends do you feel close enough to ask for help? | 4.83 | 0.38 | 1.00 |
| | 15 | When one of your friends has to make an important decision, how often does he talk about it with you? | 4.22 | 0.93 | 0.48 |

**Table 3. Social isolation and social network assessment in the round 2nd survey.**

| Category | No. | Sub-items | Mean* | SD** | CVR*** |
|---|---|---|---|---|---|
| Objective isolation | 1 | How many family members or housemates live with you? | 4.78 | 0.41 | 1.00 |
| | 2 | What public transportation is available where you currently live? | 4.00 | 1.10 | 0.39 |
| | 3 | Please select whether public transportation is conveniently accessible from your home. | 3.78 | 1.14 | 0.39 |
| | 4 | Please indicate how many times you go out of the house on average per day. | 4.61 | 0.77 | 0.83 |
| | 5 | Do you have acquaintances who visit you every day, and if so, how many? | 4.30 | 0.75 | 0.83 |
| | 6 | How many people call you or ask how you are every day? | 4.57 | 0.77 | 0.83 |
| | 7 | Are you currently communicating through Social Network Sites? | 4.09 | 0.65 | 0.73 |
| Subjective isolation | 1 | Over the past month, I've really missed my close friend. | 4.04 | 1.00 | 0.39 |
| | 2 | Over the past month, I have felt a general sense of emptiness. | 4.17 | 1.01 | 0.48 |
| | 3 | Over the past month, I've felt like my circle of friends and acquaintances was too limited. | 3.91 | 1.02 | 0.39 |
| | 4 | Over the past month, I have felt separated from others. | 4.48 | 0.83 | 0.74 |
| | 5 | Over the past month, I have felt isolated from other people. | 4.43 | 0.82 | 0.74 |
| | 6 | For the past month, I have felt lonely and friendless. | 4.48 | 0.71 | 0.74 |
| | 7 | For the past month, I've felt like I had no one to rely on. | 4.52 | 0.65 | 0.83 |
| | 8 | Over the past month, I've thought to myself that there are enough people around me that I feel close to. | 4.61 | 0.64 | 0.83 |
| | 9 | Over the past month, I have felt "in tune" with the people around me. | 4.04 | 1.04 | 0.48 |
| Social network | 1 | How many family members and relatives do you see or hear from more than once a month? | 4.30 | 1.08 | 0.57 |
| | 2 | How often do you see or hear from the family members and relatives with whom you have the most contact? | 4.04 | 1.00 | 0.39 |
| | 3 | How many of your family and relatives do you feel comfortable enough to talk to about personal matters? | 4.52 | 0.65 | 0.83 |
| | 4 | How many family members and relatives do you feel close enough to ask for help? | 4.70 | 0.55 | 0.91 |
| | 5 | When one of your family members and relatives has to make an important decision, how often does he or she talk about it with you? | 4.09 | 1.10 | 0.39 |
| | 6 | Is there always someone you can talk to about everyday problems? | 4.74 | 0.61 | 0.83 |
| | 7 | Are there a lot of people you can turn to when you have a problem? | 4.78 | 0.51 | 0.91 |
| | 8 | Are there many people I can completely trust? | 4.57 | 0.71 | 0.74 |
| | 9 | Do I have friends I can call whenever I need them? | 4.48 | 0.71 | 0.74 |
| | 10 | How many friends do you see or hear from more than once a month? | 4.52 | 0.83 | 0.74 |
| | 11 | How often do you see or hear from the friends you have the most contact with? | 3.96 | 1.12 | 0.48 |
| | 12 | How many friends do you have with whom you feel comfortable enough to talk about personal matters? | 4.78 | 0.41 | 1.00 |
| | 13 | How many friends do you feel close enough to ask for help? | 4.83 | 0.38 | 1.00 |
| | 14 | When one of your friends has to make an important decision, how often does he talk about it with you? | 4.22 | 0.93 | 0.48 |

**Table 4. Average of the expert panels.**

| | Ma | SDb | CVRc | Consensusd | Convergencee |
|---|---|---|---|---|---|
| 1st Delphi | 4.34 | 0.82 | 0.64 | 0.57 | 0.74 |
| 2nd Delphi | 4.54 | 0.57 | 0.87 | 0.31 | 0.87 |

aMean: average values of all sub-items;

bSD: standard deviation of all sub-items;

cCVR: content validity ratio of all sub-items;

dConsensus: consensus ratio of all sub-items;

eConvergence: convergence ratio of all sub-items.

down into complex and multidimensional factors [23]. For example, mobility is another important factor [24]. The World Health Organization (WHO) [25] recognized transportation as a key factor that may underscore an older adults' ability to participate in social and civic activities and health care. Furthermore, according to Levasseur et al. [26], there is an association between the built environment, including public transportation, and social participation, and public transportation is one of many factors hindering social participation. However, there is a lack of information about how public transportation is associated with social isolation [25,27,28]. Therefore, considering access to or the use of public transportation is necessary to measure older adults' social isolation. In addition, the experts proposed that to measure social isolation, a question regarding the usage of SNSs should be added, because Social Network Serveics (SNSs) are now being widely adopted by older adults [29]. Older adults primarily use SNSs to keep in touch with their family and close friends [30,31]. In terms of the item regarding employment status showed a low CVR value of 0.30 in the first Delphi survey. Some previous research demonstrated that social isolation and network are narrow in older adults especially after retirement [32,33]. However, majority of expert in this study proposed that the older adults which the main subject of the SISN assessment might be retired, thus the question was not appropriate therefore the question was removed.

Subjective isolation can be measured as a component of self-perceived isolation [34]. Based on previous research, objective isolation does not correlate with subjective isolation, or the correlation is low [35]. Thus, people with fewer contacts do not necessarily feel lonely or isolated, whereas having many social contacts does not preclude a sense of isolation [34,36]. Therefore, to accurately measure older adults' social isolation, it is important to distinguish between objective and subjective dimensions. Based on the previous research, 10 items were in initially developed to measure subjective social isolation. During the first round of the Delphi process, the expert panel reached consensus on a total 9 items, after determining that the item measuring feelings of loneliness in the presence of others was redundant with existing item. In this study, subjective social isolation, or perceived isolation, was developed ten items. The experts panel in our study supported this perspective from previous literature, confirming the importance of distinguishing between objective and subjective dimensions of social isolation in assessment tools.

Social networks are valuable aspects of life and contribute to better mental and physical health [37]. However, the measurement of social networks has been poorly standardized because of disagreements over their definition and theoretical basis [38]. In the assessment of social network items, expert consensus indicated that all items were valid, with the exception of one redundant item within the family network dimension. The experts' perspectives demonstrated consistency with previous researching findings, supporting the crucial role of social network assessment in understanding social isolation. We employed the definition of social networks provided by Berkman et al. [39]. A social network is a quantifiable relationship among individuals, families, groups, or corporations held together by a common interest, goal, or need [37,39] and facilitates engagement in social activities and promotes access to social support [39]. Recent research has increasingly concentrated on quantifying social networks, which is a more objective measure of the structural relationship and is more appropriate for understanding its association with critical health outcomes, such as cognitive function and other health

statuses [40]. Thus, as a results of this study, a total of 30 items were validated by the multidimensional expert panel: 7 items for objective isolation, 9 items for subjective isolation, and 14 items for social network assessment.

This study had several strengths. First, we conducted a comprehensive literature review to identify the most appropriate factors for measuring social isolation and networks. Second, based on the results of a previous literature review, we utilized the Delphi method, which is a widely recognized and rigorous methodology for clinical consensus in research. This enabled us to capture the opinions of experts in various fields. Third, this study is significant because the new SISN assessment tool was developed to reflect the newly changed social environment and culture of modern society, where isolation is becoming increasingly prevalent.

However, the study had some limitations. We analyzed only the content validity of the items in social isolation and networks using an expert panel. Therefore, validity and reliability studies should be conducted in the future. Additionally, only healthcare professionals from South Korea were invited to participate; thus, our panel of experts may not be generalizable to other populations.

## Conclusion

The modified Delphi method was employed to reach a consensus on the important aspects to consider when measuring the social isolation and networks of older adults. Through this process, 30 items divided into three categories were agreed upon. The SISN tool may help to clarify older adults' social isolation and networks and to identify individuals who require more in-depth social-health modification and related interventions. However, individuals' social isolation and networks tend to be affected by various environmental, geographic, and cultural factors. Further research is required to confirm the validity and reliability of this assessment.

## Supporting information

**S1 Text. SISN questionnaire.** The questionnaire contains all items of SISN assessment.
(DOCX)

## Author contributions

**Conceptualization:** Kang-Hyun Park.

**Data curation:** Ah-Ram Kim, Kang-Hyun Park.

**Formal analysis:** Ah-Ram Kim, Kang-Hyun Park.

**Funding acquisition:** Kang-Hyun Park.

**Methodology:** Ah-Ram Kim.

**Visualization:** Ah-Ram Kim.

**Writing – original draft:** Ah-Ram Kim, Kang-Hyun Park.

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
