## [Decision Letter · Decision Letter 0]

25 Feb 2025

PONE-D-25-00848Development of Social Isolation and Social Network Assessment Tool for Older Adults: A Delphi surveyPLOS ONE

Dear Dr. Park,

Thank you for submitting your manuscript to PLOS ONE. After careful consideration, we feel that it has merit but does not fully meet PLOS ONE’s publication criteria as it currently stands. Therefore, we invite you to submit a revised version of the manuscript that addresses the points raised during the review process.

We look forward to receiving your revised manuscript.

Kind regards,

Henri Tilga, PhD

Academic Editor

PLOS ONE

Journal Requirements:

“This work was supported by the National Research Foundation of Korea grant funded by the Korean Government (RS-2023-00213828).”

“This work was supported by the National Research Foundation of Korea grant funded by the Korean Government (RS-2023-00213828).”

“This work was supported by the National Research Foundation of Korea grant funded by the Korean Government (RS-2023-00213828).”

6. Please include captions for your Supporting Information files at the end of your manuscript, and update any in-text citations to match accordingly. Please see our Supporting Information guidelines for more information: http://journals.plos.org/plosone/s/supporting-information .

Reviewers' comments:

Reviewer's Responses to Questions

**Comments to the Author**

1. Is the manuscript technically sound, and do the data support the conclusions?

Reviewer #1: Yes

Reviewer #2: Yes

Reviewer #3: Yes

2. Has the statistical analysis been performed appropriately and rigorously? 

Reviewer #1: Yes

Reviewer #2: I Don't Know

Reviewer #3: Yes

3. Have the authors made all data underlying the findings in their manuscript fully available?

Reviewer #1: Yes

Reviewer #2: Yes

Reviewer #3: Yes

4. Is the manuscript presented in an intelligible fashion and written in standard English?

Reviewer #1: Yes

Reviewer #2: Yes

Reviewer #3: Yes

5. Review Comments to the Author

Reviewer #1: Thank you for submitting your paper to the Journal of Plos One. I encourage the authors to revise the manuscript in light of the comments provided below:

Methodology

Organize the methodology headings in this order:

1. Study design and participants (merge the two parts including research design and expert panel).

2. Data gathering (stages of Delphi method)

3. Data analysis

4. Ethical considerations

Transfer all ethical considerations (line 97, 102) to the ethics section at the end of methods section.

In line 105 please declare the payment provided was an incentive for participating in the study.

Reviewer #2: Thank you for the opportunity to review this paper on an important topic particularly for public health professionals.

This was an interesting article although there were several limitations to the study as it was reported in this paper and a number of omissions that may have provided further context. As previous reviews have pointed out, social isolation for older adults is a major public health concern and therefore this study should be of importance in assessing risk. This would add to the evidence that is seeking to address the gap between research and practice. However, there are important contextual aspects to social isolation and networks that have been omitted from the current study that limit its impact. The authors rightly point out that the study may not be generalisable to other cultures though they do not substantiate this with any specific reasons. The comments below are intended to help the authors review and refine their paper.

Throughout the paper, the authors refer to ‘older adults’ but do not define the age group that is being described or the specific cohort the assessment tool is targeted at or whether it is intended be generic. In the literature on the topic of social isolation and social networking, some authors refer to the over 50’s as older adults, others refer to the over 60’s and other refer to older, older adults i.e. over 70’s. It would be helpful if the authors were to clarify the age range of the population being addressed in their study as the levels of isolation may be quite different for different cohorts of older people. Also, the retirement age and subsequent loss of economic activity as well as provision for later years may vary in different countries.

The introduction did not mention the impact of the lack of finance due to the loss of economic activity, socio-economic factors, or the loss of mobility, in terms of social isolation and networks.

The authors point to some limitations of the study and should also include the omission of older adults and those with lived experience from the development of the assessment tool. I could not find any reference to the involvement of older adults so concluded they were not amongst the experts invited to participate in the study. It appears that the experts invited to take part in the Delphi process all had expertise in research or a clinical field related to social isolation and networks. Whether they had relevant experience of working with older adults is not clear. Omitting the views and expertise of those with lived experience i.e. older people is a missed opportunity and this raises concerns about the relevance of the items that were included in the assessment tool to their experience. Alternatively, it may have been possible to separately seek the views of older adults and triangulate the responses of the ‘experts’ with those with lived experience.

It was surprising that the assessment tool did not include any questions on a person’s mobility as this surely must have a major impact on their ability to network and their subsequent isolation. Similarly, whether a person lives in an urban or rural area impacts on their ability to socialise with peers or families. Other questions that appear to have been omitted but would provide context for a person’s level of isolation are whether they are resident in a community or in residential provision e.g. a care home; whether they need support or help to meet others; whether finance is an issue; whether the person is connected to others through IT or social media. Older adults may also be carers, and this can severely inhibit socialising and networking. Whilst recognising that the authors took items from existing assessment tools to utilise in their Delphi process, these areas provide valuable context for any future assessment tool.

Reviewer #3: Thank you for the opportunity to review the paper titled ‘Development of Social Isolation and Social Network Assessment Tool for Older Adults: A Delphi survey’. It is a valuable agenda in an aging society, but some of the contents need to be revised as follows:

1. As the aging society accelerates and the management of the elderly is emerging as a social issue, the development of a scale that can assess social isolation and networks is a timely study. The research problem is clearly presented in the introduction.

2. In the research method section, please add exclusion criteria as well as selection criteria when recruiting an expert panel.

I think that adding theoretical background related to social isolation and networks of the elderly will further emphasize the validity of the study to the readers.

3. In the results section, please present the full names of ‘SD’ and ‘CVR’ in Tables 2 and 3 at the bottom of the table.

4. In the discussion section, I hope to add strong significance that can act as a clinical strength by using the ‘Social Isolation and Network Assessment Tool for Older Adults’ developed in this study. In addition, please add the limitations of the study and specifically suggest future research directions.

In addition, please more specifically present the advantages and limitations of the newly developed assessment tool compared to the existing assessment tools presented as research questions.

6. PLOS authors have the option to publish the peer review history of their article (what does this mean? ). If published, this will include your full peer review and any attached files.

**Do you want your identity to be public for this peer review?** For information about this choice, including consent withdrawal, please see our Privacy Policy .

Reviewer #1: **Yes: ** Mina Hashemiparast

Reviewer #2: **Yes: ** Dr Virginia Minogue

Reviewer #3: No

---

## [Author Response · Author response to Decision Letter 1]

19 Mar 2025

Editors

PLOS ONE

March, 17, 2025

Dear Editor,

We sincerely thank you and the reviewers for your thorough examination and insightful critiques of our manuscript. We are particularly grateful for the constructive feedback provided, which has significantly contributed to enhancing the quality and scientific rigor of our work. We appreciate the opportunity to revise and resubmit our manuscript for further consideration by the editorial board of PLOS ONE.

Herewith, we are pleased to present the revised version of our research article, which has been substantially improved in accordance with the valuable suggestions received during the review process. We have diligently addressed each point raised and have implemented appropriate modifications throughout the manuscript to strengthen its scientific merit, methodological clarity, and overall presentation.

Below, please find our comprehensive point-by-point response addressing each concern and recommendation put forth by the academic editor and reviewers. For clarity, we have highlighted the changes made in the revised manuscript and provided detailed explanations justifying our approach to each modification. We believe that we have thoroughly and satisfactorily addressed all the issues identified during the review process, resulting in a significantly enhanced contribution to the field.

Regarding the data availability requirements, we confirm that all necessary data to replicate our findings are included within the manuscript and its Supporting Information files, as specified in our Data Availability Statement.

On behalf of my co-authors, I express our profound appreciation for your consideration of this revised submission. We value the rigorous peer-review process facilitated by PLOS ONE and believe it has substantially improved the quality of our work. We look forward to your evaluation and subsequent decision regarding our manuscript.

Yours sincerely,

Park, Kang-Hyun, Ph.D. (Corresponding author)

kanghyun@bu.ac.ke

---

## [Decision Letter · Decision Letter 1]

4 Apr 2025

Development of Social Isolation and Social Network Assessment Tool for Older Adults: A Delphi survey

PONE-D-25-00848R1

Dear Dr. Park,

We’re pleased to inform you that your manuscript has been judged scientifically suitable for publication and will be formally accepted for publication once it meets all outstanding technical requirements.

Kind regards,

Henri Tilga, PhD

Academic Editor

PLOS ONE

Additional Editor Comments (optional):

Reviewers' comments:

Reviewer's Responses to Questions

**Comments to the Author**

1. If the authors have adequately addressed your comments raised in a previous round of review and you feel that this manuscript is now acceptable for publication, you may indicate that here to bypass the “Comments to the Author” section, enter your conflict of interest statement in the “Confidential to Editor” section, and submit your "Accept" recommendation.

Reviewer #2: All comments have been addressed

Reviewer #3: All comments have been addressed

2. Is the manuscript technically sound, and do the data support the conclusions?

Reviewer #2: Yes

Reviewer #3: Yes

3. Has the statistical analysis been performed appropriately and rigorously? 

Reviewer #2: I Don't Know

Reviewer #3: Yes

4. Have the authors made all data underlying the findings in their manuscript fully available?

Reviewer #2: Yes

Reviewer #3: Yes

5. Is the manuscript presented in an intelligible fashion and written in standard English?

Reviewer #2: Yes

Reviewer #3: Yes

6. Review Comments to the Author

Reviewer #2: I am satisfied that my comments have been addressed. I recommend the authors undertake further proof reading to ensure they eliminate any remaining typographical and grammatical errors particularly those that may have arisen during the re-submission process.

Reviewer #3: The authors congratulate the authors for revising and upgrading the manuscript. I believe that this manuscript sufficiently reflects the reviewers' comments and is publishable.

7. PLOS authors have the option to publish the peer review history of their article (what does this mean? ). If published, this will include your full peer review and any attached files.

**Do you want your identity to be public for this peer review?** For information about this choice, including consent withdrawal, please see our Privacy Policy .

Reviewer #2: **Yes: ** Dr Virginia Minogue

Reviewer #3: No

---

## [Editor Report · Acceptance letter]

PONE-D-25-00848R1

PLOS ONE

Dear Dr. Park,

I'm pleased to inform you that your manuscript has been deemed suitable for publication in PLOS ONE. Congratulations! Your manuscript is now being handed over to our production team.

Kind regards,

on behalf of

Dr. Henri Tilga

Academic Editor

PLOS ONE